# SHAPE ASSEMBLY VIA EQUIVARIANT DIFFUSION

## ABSTRACT

We tackle the problem of solving shape puzzles, that is, reassembling randomly-partitioned and scattered pieces of 2D or 3D shapes into an original shape. This task is challenging since it only relies on geometric features without rich visual information. Specifically, we are supposed that target shapes and their randomly-partitioned pieces are pattern-free and irregular. Existing methods tend to rely on specific constraints regarding piece shapes and neglect the consideration of invariance and equivariance. We propose learning a robust puzzle solver through a generative diffusion process in which the roto-translational equivariance holds. Experiments on 2D and 3D puzzle benchmarks including the Breaking Bad dataset demonstrate that our method successfully assembles given geometric pieces into a target shape. We also provide in-depth ablation studies showing the effects of our equivariant design and the components in our proposed framework.

## 1 INTRODUCTION

The task of *shape puzzles* aims to assemble given parts into a complete and assembled shape. In particular, we solve pattern-free and irregular shape puzzles by assuming the unavailability of visual and semantic information, unlike semantic shape assembly approaches (Li et al., 2020a;b; Lee et al., 2021; Wang et al., 2022). More specifically, the problem solved in this work more focuses on the geometric information of each shape rather than semantic information.

As discussed in the previous work (Hosseini et al., 2023) on a 2D geometric puzzle dataset, there are a few inherent characteristics of simple puzzles, making shape puzzles a highly suitable task for exploring the potential of geometric features. Firstly, the puzzle dataset is built with a minimal number of points defining each shape, simplifying a solution space. Secondly, each puzzle can possess explicit geometric constraints such as edge matching constraint. Lastly, it is easy to control the puzzle's constraints, one can adjust the level of generality within the task.

Recognizing the potential of solving the 2D puzzle dataset, our work first proposes the creation of a generic 3D puzzle dataset, and then showcases a novel puzzle solver based on a diffusion model-based conditional generation framework for 2D and 3D puzzles including the Breaking Bad dataset (Sellán et al., 2022). Inspired by the existing studies on complex representations and SE(3) equivariant network such as the work (Deng et al., 2021), our approach demonstrates performance improvements by integrating simple yet effective representations that satisfy permutation equivariance, SE(2)/SE(3) invariance, and potentially SE(2)/SE(3) equivariance. This design is theoretically and empirically validated with significant performance gains in both 2D and 3D puzzle tasks.

In summary, our contributions are as follows.

- We design a dataset generating framework which allows direct adjustment of piece number and shape complexity, and propose a new 3D puzzle dataset.

- We propose a novel diffusion-based framework which solves puzzles on 2D and 3D domains. To the best of our knowledge, this is the first approach to solving a 3D assembly task via the coniditional equivariant diffusion model.

- We demonstrate theoretical results and performance improvements with ablation studies proving the effectiveness of each component included in our framework.

*We will make our implementation public upon publication.*

## 2 RELATED WORK AND PRELIMINARIES

We cover the recent work and preliminaries related to our framework in this section. More detailed discussion on related work and preliminaries can be found in Section A.

**Shape and Puzzle Assembly Problems.** Shape assembly is an active research area in computer vision and graphics. It is to construct complete shapes from given parts. Previous research often utilizes public datasets such as PartNet (Mo et al., 2019) and IKEA-Manual (Wang et al., 2022), which provide detailed semantic part information. Thus, they can be categorized into semantic shape assembly. The recent studies (Li et al., 2020a;b; Lee et al., 2021; Wang et al., 2022) have addressed assembly from single images or focused on specific applications such as furniture assembly.

While many prior approaches heavily rely on semantic part information, the work (Cho et al., 2010; Chen et al., 2022) has tackled an assembly problem by prioritizing shape geometries over semantic details. Also, the Breaking Bad dataset (Sellán et al., 2022) has introduced challenges in assembling non-semantic fragments into complete shapes, highlighting the complexity of fractured shape reassembly. On the other hand, there is a line of research on shape assembly with LEGO bricks (Thompson et al., 2020; Kim et al., 2020; Chung et al., 2021), where each brick is semantically meaningless. Moreover, the work (Lee et al., 2022) tackles the problem of 2D sequential puzzles with fixed target shapes using the Transformer-based network.

**SE(2)/SE(3) Invariance and Equivariance.** The special Euclidean groups in two and three dimensions (denoted as the SE(2) and SE(3) groups, respectively) represent a set of all possible combinations of rotation and translation in 2D and 3D spaces. For the sake of brevity, *we only describe the details of the SE(3) group in this section*; the SE(3) group can readily be reduced to the SE(2) group. Each instance of the SE(3) group is represented by a tuple of a rotation matrix $\mathbf{R} \in \mathbb{R}^{3 \times 3}$ and a translation vector $\mathbf{t} \in \mathbb{R}^3$, i.e., $(\mathbf{R}, \mathbf{t})$. Specifically, $\mathbf{R}$ indicates an orientation change in 3D space where this matrix is orthogonal; $\mathbf{R}^\top \mathbf{R} = \mathbf{I}$, and has a determinant of 1, ensuring that it represents pure rotation without reflection. Moreover, $\mathbf{t}$ indicates a position shift in 3D space where it specifies how much an object moves along $x$, $y$, and $z$ directions. Formerly, a function $f$ to apply rotation and translation to an input variable $\mathbf{x}$ is given by the following: $f(\mathbf{x}; (\mathbf{R}, \mathbf{t})) = \mathbf{R}\mathbf{x} + \mathbf{t} \in \mathbb{R}^3$.

With the definition of the SE(3) group, SE(3) invariance and equivariance are significant for defining robustness against arbitrary SE(3) transformations, supposing that a function $f$ invariant or equivariant against a transformation $T$ if it satisfies $f(T(\mathbf{x})) = f(\mathbf{x})$ or $f(T(\mathbf{x})) = T(f(\mathbf{x}))$, respectively. These properties are particularly desirable in specific tasks where the tasks should not depend on transformations such as scaling, rotation, or translation of an input data. Without loss of generality, we can define the SE(3) invariant function $f$: $f(\mathbf{R}\mathbf{x} + \mathbf{t}) = f(\mathbf{x}) \ \forall (\mathbf{R}, \mathbf{t}) \in$ SE(3). This invariance property implies that the outputs of $f$ are all identical when applying any SE(3) transformation, so that it ensures that a model is invariant to object's positioning and orientation. Furthermore, the SE(3) equivariance is readily defined, based on the equivariance and SE(3) group. These are well-known desirable properties in robotics and 3D vision (Chen et al., 2021; Thomas et al., 2018).

## 3 PROBLEM STATEMENT

We describe the details of the 2D and 3D shape puzzle problems, focusing on the 3D puzzle task.

### 3.1 GEOMETRIC SHAPE PUZZLE TASKS

The task of 2D and 3D shape puzzles involves solving the reconstruction of irregular 2D and 3D objects from disassembled, irregular, and pattern-free pieces, as shown in Figure 1. The goal of this task is to accurately identify the correct spatial orientation and proper placement of all geometric pieces. In particular, irregular target objects and geometric pieces complicate the assembly task as it requires the precise understanding of spatial relationships between pieces and demands complex reasoning on the occupation of 2D and 3D spaces. Moreover, the characteristics on pattern-free objects make the problems more difficult since we cannot resort to the visual information captured by modern feature extraction techniques. To address such tasks, in this paper we employ the equivariant diffusion model with overcomplete representations; see Section 4 for details.

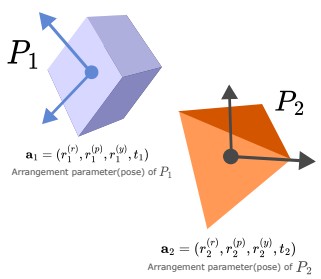
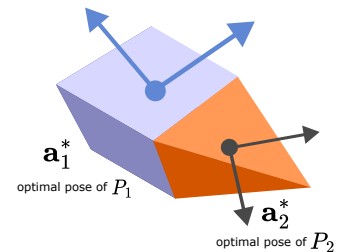

(a) Two randomly-partitioned pieces

(b) Target assembled shape

Figure 1: Given a variable number of randomly-partitioned pieces, the goal of a shape puzzle is to determine the correct positions and orientations of these pieces.

## 3.2 REPRESENTATIONS FOR 3D SHAPE PUZZLES

Suppose that we are given a puzzle set with $n$ randomly-partitioned pieces, denoted as $\mathcal{S} = \{\mathcal{P}_1, \mathcal{P}_2, \ldots, \mathcal{P}_n\}$. Each piece $\mathcal{P}_i$ is composed of an arbitrary number of *corners* and an arbitrary number of *faces*: $\mathcal{P}_i = (\mathcal{C}_i, \mathcal{F}_i)$, where $\mathcal{C}_i = \{\mathbf{c}_{i,1}, \mathbf{c}_{i,2}, \ldots, \mathbf{c}_{i,m_i}\}$ and

$\mathcal{F}_i = \{\mathbf{f}_{i,1}, \mathbf{f}_{i,2}, \ldots, \mathbf{f}_{i,l_i}\}$. Note that $\mathbf{c}_{i,j} \in \mathbb{R}^3$ is the $j$-th corner coordinate of a piece $\mathcal{P}_i$ and $\mathbf{f}_{i,k}$ is a set containing the indices which form the $k$-th face of $\mathcal{P}_i$. For example, $\mathbf{f}_{i,k} = \{1, 4, 7\}$ indicates that the $k$-th face of $\mathcal{P}_i$ consists of $\mathbf{c}_{i,1}$, $\mathbf{c}_{i,4}$, and $\mathbf{c}_{i,7}$. For $\mathcal{S}$, we define the following geometric parameters $\mathbf{a}_i = (\mathbf{r}_i^{(r)}, \mathbf{r}_i^{(p)}, \mathbf{r}_i^{(y)}, \mathbf{t}_i) \in [-1, 1]^6 \times \mathbb{R}^3$, which is called *arrangement parameter* for each $\mathcal{P}_i$. This arrangement parameter is used to place the piece in an absolute 3D coordinate system where $\mathbf{r}_i^{(r)}, \mathbf{r}_i^{(p)}, \mathbf{r}_i^{(y)} \in [-1, 1]^2$ are transformed angles for roll, pitch, and yaw directions; each angle is considered as representing $\mathbf{r}_i = [\cos(\theta_i), -\sin(\theta_i)]$, and $\mathbf{t}_i \in \mathbb{R}^3$ is a piece center. Given the specific $\mathbf{a}_i$, the transformed $\widehat{\mathbf{c}}_{i,j}$ of $\mathcal{P}_i$ is expressed as the following:

$$\widehat{\mathbf{c}}_{i,j} = \mathbf{R}_i^{(y)} \mathbf{R}_i^{(p)} \mathbf{R}_i^{(r)} \mathbf{c}_{i,j} + \mathbf{t}_i, \tag{1}$$

where rotation matrices $\mathbf{R}_i^{(r)}, \mathbf{R}_i^{(p)}, \mathbf{R}_i^{(y)}$ are defined as follows:

$$\mathbf{R}_i^{(r)} = \begin{bmatrix} 1 & 0 & 0 \\ 0 & [\mathbf{r}_i^{(r)}]_1 & [\mathbf{r}_i^{(r)}]_2 \\ 0 & -[\mathbf{r}_i^{(r)}]_2 & [\mathbf{r}_i^{(r)}]_1 \end{bmatrix}, \mathbf{R}_i^{(p)} = \begin{bmatrix} [\mathbf{r}_i^{(p)}]_1 & 0 & -[\mathbf{r}_i^{(p)}]_2 \\ 0 & 1 & 0 \\ [\mathbf{r}_i^{(p)}]_2 & 0 & [\mathbf{r}_i^{(p)}]_1 \end{bmatrix}, \mathbf{R}_i^{(y)} = \begin{bmatrix} [\mathbf{r}_i^{(y)}]_1 & [\mathbf{r}_i^{(y)}]_2 & 0 \\ -[\mathbf{r}_i^{(y)}]_2 & [\mathbf{r}_i^{(y)}]_1 & 0 \\ 0 & 0 & 1 \end{bmatrix}. \tag{2}$$

Note that $[\mathbf{r}]_k$ indicates the $k$-th entry of $\mathbf{r}$. Eventually, the goal of 3D shape puzzle task is converted to predict appropriate $\mathbf{a}_1, \mathbf{a}_2, \ldots, \mathbf{a}_n$ given $\mathcal{S}$. It is important to acknowledge that there exist multiple solutions for $\mathbf{a}_1, \mathbf{a}_2, \ldots, \mathbf{a}_n$ due to a vast number of possibilities to assemble given geometric pieces into a target shape under the SE(3) transformations. This nature complicates the learning process for addressing the puzzle problems.

## 4 PROPOSED METHOD

Our framework with the diffusion models (Ho et al., 2020) based on the Transformer network (Vaswani et al., 2017) and inspired by the previous puzzle-solving method (Hosseini et al., 2023) aims to predict piece centers and their rotation vectors. For the simplicity of the paper structure, *the description of our method is based on 3D shape puzzles*; without loss of generality, 3D puzzles can easily be reduced to 2D puzzles.

### 4.1 OVERCOMPLETE REPRESENTATIONS FOR 3D SHAPE PUZZLES

As presented in Section 3.2, we formulate the 3D shape puzzle problem as a task that estimates arrangement parameters $\mathbf{a}_1, \mathbf{a}_2, \ldots, \mathbf{a}_n$ given $\mathcal{S}$. Instead of directly predicting $\mathbf{a}_1, \mathbf{a}_2, \ldots, \mathbf{a}_n$, we define *overcomplete representations* $\mathbf{a}_{i,j}$ to utilize the information of each $\mathbf{c}_j$ of $\mathcal{P}_i$. Subsequently,

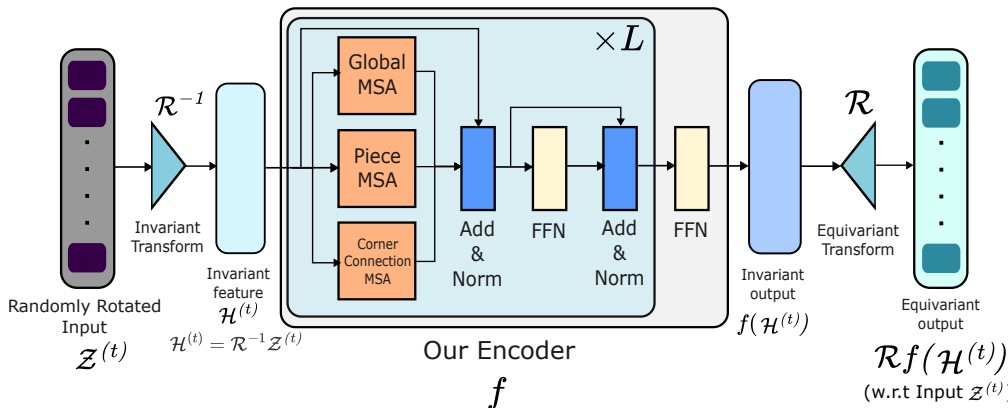

Figure 2: Overall architecture of our proposed model.

during an inference phase, we calculate the arithmetic mean of the predictions of overcomplete representations, which is used to indirectly estimate $\mathbf{a}_i$: $\mathbf{a}_i = \frac{1}{m_i} \sum_{j=1}^{m_i} \mathbf{a}_{i,j}$. By the definition of overcomplete representations, our ultimate goal is to predict $\mathbf{a}_i$ for $i \in [n]$ utilizing the corner-wise arrangement parameters $\mathbf{a}_{i,1}, \mathbf{a}_{i,2}, \ldots, \mathbf{a}_{i,m_i}$ for $i \in [n]$. Moreover, given the ground-truth arrangement parameter $\mathbf{a}_i^*$, each ground-truth corner-wise arrangement parameter $\mathbf{a}_{i,j}^*$ for $\mathcal{P}_i$ is naturally identical to $\mathbf{a}_i^*$.

While the direct estimation of all $\mathbf{a}_i$ seems simple, this indirect approach to predicting all $\mathbf{a}_{i,j}$ has proved efficient in the recent conditional generation task (Shabani et al., 2023). We presume that it naturally allows for more complex shapes to have longer queries, thereby facilitating more complex reasoning, and each corner $\mathbf{c}_{i,j}$ can be individually encoded into a fixed-length conditional vector that corresponds to $\mathbf{a}_{i,j}$ instead of the use of the piece-wise aggregated representation.

## 4.2 NEURAL ARCHITECTURE OF OUR PROPOSED METHOD

**Diffusion Models.** Following the common process of the diffusion model discussed in Section A, $\mathbf{a}_{i,j}$ is transformed into random noise during the forward process. Conversely, in the reverse process, the reconstruction model $\epsilon_\theta(\mathcal{Z}^{(t)}, t)$ generates cleaner $\mathbf{a}_{i,1}^{(t-1)}, \mathbf{a}_{i,2}^{(t-1)}, \ldots, \mathbf{a}_{i,m_i}^{(t-1)}$ for $i \in [n]$, taking $\mathcal{Z}^{(t)}$:

$$\mathcal{Z}^{(t)} = \left[ \mathbf{z}_{i,1}^{(t)}, \mathbf{z}_{i,2}^{(t)}, \ldots, \mathbf{z}_{i,m_i}^{(t)} \right]_{i=1}^{n}, \tag{3}$$

where $\mathbf{z}_{i,j}^{(t)} = [\mathbf{a}_{i,j}^{(t)}, \mathbf{c}_{i,j}, i]$ with a noisier arrangement parameter $\mathbf{a}_{i,j}^{(t)}$, the $j$-th corner $\mathbf{c}_{i,j}$, and a piece index $i$. We collect $\mathbf{z}_{i,j}^{(t)}$ in the order of $i, j$ to make $\mathcal{Z}^{(t)}$ sequential, so that we can make use of the Transformer encoder (Vaswani et al., 2017) for $\epsilon_\theta(\mathcal{Z}^{(t)}, t)$. Note that $\mathbf{a}_{i,j}^{(t)}$ only depends on $t$, while $\mathbf{c}_{i,j}$ and $i$ remain identical during the reverse process. Also, our approach employs the cosine scheduling (Nichol & Dhariwal, 2021) to transform an input data to unbiased noise in the forward process.

**Transformer-based Networks.** The Transformer network (Vaswani et al., 2017) is widely used in natural language processing and its applications (Devlin et al., 2018; Brown et al., 2020; OpenAI, 2023). However, it is inherently permutation-sensitive due to positional embedding over inputs. Inspired by the work (Zaheer et al., 2017; Lee et al., 2019), we utilize the Transformer network without positional embedding in our problem formulation, so that our network is not sensitive for the permutation of indices $i, j$ for the overcomplete representations of shape puzzles.

As illustrated in Figure 2, our model encompasses three distinctive modules of masked multi-head attention: *global multi-head self-attention*, *piece-wise masked multi-head self-attention*, and *connection-wise masked multi-head self-attention*. Firstly, the global attention mechanism employs conventional self-attention, allowing every query to attend to each other universally. This component is capable of facilitating comprehensive context understanding across entire inputs. Secondly, the piece-wise attention separately applies self-attention piece-by-piece by masking out interactions

between corners that belong to different pieces. Lastly, the connection-wise attention is capable of processing piece information $\mathcal{P}_i$ for $i \in [n]$, allowing us to directly capture relationships between connected corners. This design helps understand our puzzle problems from the macroscopic and microscopic perspectives including global layout comprehension and intricate piece-wise and connection-specific details. We conduct ablation studies on the two attention modules against the global attention in Section 6.

## 4.3 EQUIVARIANT FEATURE EMBEDDING

We introduce a feature embedding layer for our framework on shape puzzles. This layer embeds the concatenated inputs $\mathcal{Z}^{(t)}$, which are mentioned in Section 4.2. In the sense of puzzle solving, it is considered as satisfying the invariant property, but our model can be viewed as an equivariant framework in the perspective of an absolute coordinate system. In this step, our model processes various input variables under the presence of permutation and global SE(3) transformations. Our embedding layer is defined as the following:

$$h(\mathbf{z}_{i,j}^{(t)}, t) = \mathrm{lin}_i(f_{\mathrm{ac}}(\mathbf{a}_{i,j}^{(t)})) + \mathrm{lin}_c([f_{\mathrm{sc}}(\mathbf{c}_{i,j}; \mathcal{C}_i), f_{\mathrm{cc}}(f_{\mathrm{sc}}(\mathbf{c}_{i,j}; \mathcal{C}_i)), \mathrm{anchor}(i)]) + \mathrm{te}(t), \qquad (4)$$

where $\mathbf{z}_{i,j}^{(t)} = [\mathbf{a}_{i,j}^{(t)}, \mathbf{c}_{i,j}, i]$ is an enhanced input for $\mathbf{c}_{i,j}$ of $\mathcal{P}_i$, $\mathrm{anchor}(\cdot)$ is a function to distinguish an anchor piece, $\mathrm{lin}_i(\cdot)$ and $\mathrm{lin}_c(\cdot)$ are simple linear functions with learnable parameters; in this paper we embed input vectors to 512-dimensional vectors, and $\mathrm{te}(\cdot)$ is a conventional time embedding function, which is used in the work (Ho et al., 2020). From now, we detail the respective components presented in (4). Similar to (3), we can define the collection of (4) for $i \in [n]$:

$$\mathcal{H}^{(t)} = \left[ h(\mathbf{z}_{i,1}^{(t)}, t), h(\mathbf{z}_{i,2}^{(t)}, t), \dots, h(\mathbf{z}_{i,m_i}^{(t)}, t) \right]_{i=1}^{n}. \qquad (5)$$

**Shape Canonicalization $f_{\mathrm{sc}}$.** In the prior work, the positional information of each corner, denoted as $\mathbf{c}_{i,j}$, is significant for encoding the shape of each geometric piece. Despite its importance, the use of this information makes the problems ambiguous. In particular, identical shapes can be encoded differently. For example, given $(\mathbf{R}, \mathbf{t}) \in \mathrm{SE}(3)$, $\mathcal{P}_i = (\mathcal{C}_i, \mathcal{F}_i)$ and $\mathcal{P}_i' = ((\mathbf{R}, \mathbf{t})\mathcal{C}_i, \mathcal{F}_i)$ clearly indicate topologically equal shapes, but they are considered as different shapes in the prior work. Note that $(\mathbf{R}, \mathbf{t})\mathcal{C}_i = \{\mathbf{R}\mathbf{c}_{i,1} + \mathbf{t}, \mathbf{R}\mathbf{c}_{i,2} + \mathbf{t}, \dots, \mathbf{R}\mathbf{c}_{i,m_i} + \mathbf{t}\}$. Instead of applying a complex SE(3) invariant model that significantly restricts a representation space, we propose a simple SE(3) invariant function $f_{\mathrm{sc}} : \mathbb{R}^3 \to \mathbb{R}^3$, which effectively standardizes corner coordinates to make the regular representation of shapes. To design SE(3) invariant $f_{\mathrm{sc}}$ which satisfies the following:

$$f_{\mathrm{sc}}(\mathbf{c}_{i,j}; \mathcal{C}_i) = f_{\mathrm{sc}}(\mathbf{R}\mathbf{c}_{i,j} + \mathbf{t}; (\mathbf{R}, \mathbf{t})\mathcal{C}_i), \quad \forall (\mathbf{R}, \mathbf{t}) \in \mathrm{SE}(3), \qquad (6)$$

we eliminate the effect of arbitrary rotation and translation $(\mathbf{R}, \mathbf{t}) \in \mathrm{SE}(3)$ from $\mathbf{R}\mathbf{c}_{i,j} + \mathbf{t}$. The translation vector is computed by the center of mass as $\mathbf{t} = \mathbf{R}(\frac{1}{n}\sum_{i=1}^{n} \frac{1}{m_i} \sum_{j=1}^{m_i} \mathbf{c}_{i,j}) + \mathbf{t} = \frac{1}{n}\sum_{i=1}^{n} \frac{1}{m_i} \sum_{j=1}^{m_i} (\mathbf{R}\mathbf{c}_{i,j} + \mathbf{t})$ with the assumption of the center of $\mathbf{c}_{i,j}$ is located at the origin. After substracting the translation vector, the rotation matrix is obtained through singular value decomposition (SVD). Initially, the centered corners are sorted and arranged into a matrix $\mathbf{C}_R \in \mathbb{R}^{3 \times m_i}$ to have fixed order in the same set. This arrangement ensures that the SVD results remain consistent, even in the presence of rotational symmetry. The SVD decomposition gives us $\mathbf{C}_R = \mathbf{U}\boldsymbol{\Sigma}\mathbf{V}^\top$, where $\mathbf{U} \in \mathbb{R}^{3 \times 3}$ identifies the three principal axes that best describe the data, $\boldsymbol{\Sigma} \in \mathbb{R}^{3 \times 3}$ is the diagonal matrix of singular values, and $\mathbf{V}^\top \in \mathbb{R}^{3 \times m_i}$ contains the right singular vectors corresponding to the singular values. However, each principal axis has a sign ambiguity under different rotations, and $\mathbf{U}$ can represent a reflection with a determinant of -1. To resolve this, the sign ambiguity is eliminated by aligning the signs of $\mathbf{U}$ with the signs of the largest absolute position of $\mathbf{V}$ along each axis. The determinant of $\mathbf{U}$ is then corrected by multiplying a corresponding axis of $\mathbf{U}$ and $\mathbf{V}$ by the determinant of $\mathbf{U}$, ensuring that the overall determinant becomes 1. After this sign and determinant correction, we can obtain the desired rotation matrix $\mathbf{R} = \mathbf{US}$, where $\mathbf{S}$ represents the correction matrix. Once $\mathbf{R}$ is canceled out, the corners are left in the form of $\boldsymbol{\Sigma}(\mathbf{VS})^\top$, which is now rotation-invariant. The result, $\boldsymbol{\Sigma}\mathbf{SV}^\top$, represents the canonicalized shape, while $\mathbf{US}$ gives the rotation. The resulting translation and rotation are then incorporated back into the arrangement parameters with the relation of $\mathbf{R}_{a_i}(\mathbf{R}\mathbf{c}_{i,j} + \mathbf{t}) + \mathbf{t}_{a_i} = (\mathbf{R}_{a_i}\mathbf{R})\mathbf{c}_{i,j} + (\mathbf{R}_{a_i}\mathbf{t} + \mathbf{t}_{a_i})$, where $\mathbf{R}_{a_i}$ and $\mathbf{t}_{a_i}$ denote the corresponding SE(3) transformation of the parameter $\mathbf{a}_i$. The corners and the arrangement parameters become invariant under the arbitrary transformation of SE(3) by considering the fixed target shape.

**Anchor Centering $f_{\mathbf{ac}}$ and Anchor Selection.** We now introduce the method of an anchor centering strategy, which re-aligns all arrangement parameters $\mathbf{a}_{1,1}^{(t)}, \mathbf{a}_{1,2}^{(t)}, \ldots, \mathbf{a}_{n,m_n}^{(t)}$ at step $t$ with respect to the expected arrangement parameter from the reverse process $\mathbf{a}_{p,1}^{(t)}$ of an anchor piece $\mathcal{P}_p$. An anchor-centered new input representation $f_{\mathrm{ac}}(\mathbf{a}_{i,j}^{(t)})$ is given by the following:

$$f_{\mathrm{ac}}(\mathbf{a}_{i,j}^{(t)}) = \left(\mathbf{a}_{p,1}^{(t)}\right)^{-1} \mathbf{a}_{i,j}^{(t)}. \tag{7}$$

Through the components discussed, we transform the model's input space to be invariant to global SE(3) transformations. However, the simple application of these changes disrupts consistency of the training process of diffusion. Originally, the reconstruction model for the reverse process $\epsilon_\theta(\mathcal{Z}^{(t)}, t)$ is trained to match to the ground-truth noise $\epsilon_{i,j}^{(t)}$ of the forward process. In order to make this training process consistent, we apply reverse transformation to both target ground-truth noise and the predicted noise of the reverse process model:

$$\bar{\epsilon}_{i,j}^{(t)} = (\mathbf{a}_{p,1}^{(t)})\epsilon_{i,j}^{(t)} \quad \text{and} \quad \tilde{\epsilon}_{i,j}^{(t)} = (\mathbf{a}_{p,1}^{(t)})\hat{\epsilon}_{i,j}^{(t)}, \tag{8}$$

where $\epsilon_{i,j}^{(t)}$ is original ground-truth noise and $\hat{\epsilon}_{i,j}^{(t)}$ is an original model output. Using this transformation, our loss for training diffusion models changes accordingly; it will be discussed more in Section 4.4. Similar to (12), the model's reverse process still denoises $\mathbf{a}_{i,j}^{(t)}$ by predicting $\tilde{\epsilon}_{i,j}^{(t)}$:

$$\mathbf{a}_{i,j}^{(t-1)} = \frac{1}{\sqrt{1-\beta_t}} \left(\mathbf{a}_{i,j}^{(t)} - \frac{\beta_t}{\sqrt{1-\bar{\alpha}_t}} \tilde{\epsilon}_\theta(f_{\mathrm{ac}}(\mathbf{a}_{i,j}^{(t)}), t)\right). \tag{9}$$

To actively use the information from the anchor, we add an anchor indicator:

$$\mathrm{anchor}(i) = \begin{cases} 1, & i = p, \\ 0, & \text{otherwise}, \end{cases} \tag{10}$$

where $p$ is an anchor index. This additional indicator encourages the model to focus on the anchor.

Lastly, we consider one of two criteria for selecting an anchor $\mathcal{P}_p$: choosing either a randomly-selected piece or the largest piece as an anchor This uncovers the importance of the anchor's size in the assembly process, suggesting that larger pieces serve as a more stable and recognizable beacon for solving puzzles. This is intuitive since larger pieces are likely to have more geometric features. Furthermore, since the selection of largest piece $p$ is permutation-equivariant against $i$, this selection also makes $p$-dependent anchor$(i)$ permutation-equivariant.

**Corner Connectivity $f_{\mathbf{cc}}$.** In addition to shape canonicalization, which effectively encodes each piece shape into its inherent representation, we define additional vectors to indirectly encode relative information between connected corners in $\mathcal{P}_i$. While $\mathbf{c}_{i,j}$ can capture the information based on corner locations, it does not directly hold the surface information of the piece $\mathbf{c}_{i,j}$ belongs to.

Assuming that the given connected corners of $\mathbf{c}_{i,j}$ are $\{\mathbf{c}_{i,a}, \mathbf{c}_{i,b}, \mathbf{c}_{i,c}\}$, The output of $f_{\mathrm{cc}}(\mathbf{c}_{i,j})$ is a zero-padded 30-dimensional vector of the concatenation of all unit vectors $\frac{\mathbf{c}_{i,a}-\mathbf{c}_{i,j}}{\|\mathbf{c}_{i,a}-\mathbf{c}_{i,j}\|_2^2}$, $\frac{\mathbf{c}_{i,b}-\mathbf{c}_{i,j}}{\|\mathbf{c}_{i,b}-\mathbf{c}_{i,j}\|_2^2}$, and $\frac{\mathbf{c}_{i,c}-\mathbf{c}_{i,j}}{\|\mathbf{c}_{i,c}-\mathbf{c}_{i,j}\|_2^2}$. This configuration is also crucial from the invariance perspective. Since it is constructed based on the aforementioned canonicalized shape, it shares equal SE(3) invariance against $\mathcal{C}_i$. Furthermore, by relying on ground-truth connection information, it preserves permutation invariance with respect to the permutation of $i, j$. This ensures that the representation of our model is both comprehensive in capturing the geometric features of each piece and robust to variations in orientation and ordering.

## 4.4 LOSS FUNCTIONS

For training, we employ the mean squared error (MSE) loss on noise, widely used in diffusion models, along with a matching loss to support the puzzle-solving processes. The MSE loss is defined by $L_{\mathrm{noise}} = \sum_{(i,j)} \|\bar{\epsilon}_{i,j}^{(t)} - \tilde{\epsilon}_{i,j}^{(t)}\|_2^2$. Note that the transformed noise shown in (8) is used in the MSE loss. Additionally, we calculate the matching loss for corner matching. For example, if $j$-th corner of $\mathcal{P}_i$ matches to the $k$-th corner of $\mathcal{P}_l$ in the ground-truth puzzle set, the following matching loss is applied to the given pair: $L_{\mathrm{match}} = \sum_{(i,k,l,m)} \|(\mathbf{R}_i^{(y)}\mathbf{R}_i^{(p)}\mathbf{R}_i^{(r)}\mathbf{c}_{i,j} + \mathbf{t}_i) - (\mathbf{R}_l^{(y)}\mathbf{R}_l^{(p)}\mathbf{R}_l^{(r)}\mathbf{c}_{l,k} + \mathbf{t}_l)\|_2^2$. Eventually, we use the sum of two losses as an overall loss: $L = L_{\mathrm{noise}} + L_{\mathrm{match}}$.

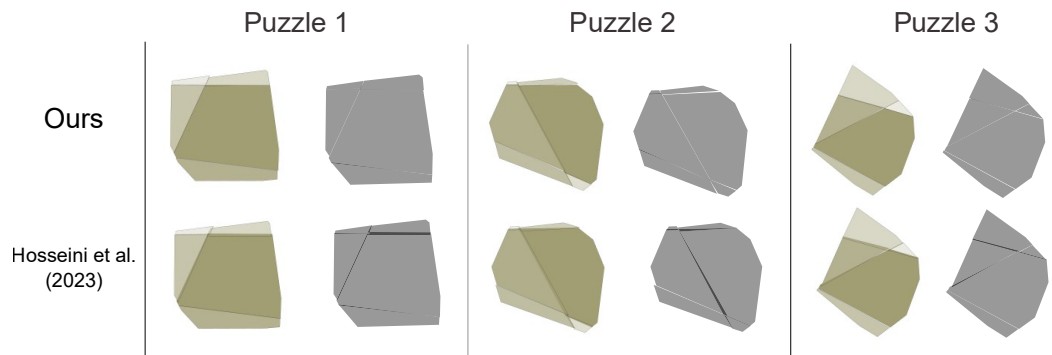

Figure 3: Qualitative results for the 2D cross-cut dataset.

## 5 THEORETICAL ANALYSIS ON OUR METHOD

We leverage equivariance to enhance the model's training efficiency and increase its robustness to transformations applied to the puzzle. Specifically, the proposed model enforces an SE(3)-invariant density function for the assembled puzzle, allowing for a more efficient assembly process. In this section, we provide the theoretical analysis to support this approach.

In order to analyze our model from the perspective of whole assembly processes, we suppose that we are interested in the invariance of a set of all pieces, each of which is defined on the SE(3) manifold. In particular, $\widehat{\mathcal{C}}_i$ is defined by multiplying the arrangement parameters to $\mathcal{C}_i$ for $i \in [n]$, as mentioned in Section 3.2. The translation part of the arrangement parameters is naturally defined on the SE(3) manifold, but their rotation part may not be on the SE(3) manifold due to our definition on orientation. This issue can be readily resolved by the normalization of the rotation part of the arrangement parameters.

**Proposition 1.** *Suppose that $\mathcal{C}_i$ is a representation without shape canonicalization. There always exists some arrangement parameter for transforming $\mathcal{C}_i$ to the canonicalized $\mathcal{C}_i$.*

**Proposition 2.** *Let $\widehat{\mathcal{C}}$ be an assembled puzzle of pieces $\mathcal{C}$ with arrangement parameters. By using the arrangement parameters sampled from the standard Gaussian distributions, $\widehat{\mathcal{C}}_i$ is a linear transformation of the canonicalized $\mathcal{C}_i$ and the sampled arrangement parameter for $i \in [n]$. Therefore, the consideration of the sampled arrangement parameter for calculating $\widehat{\mathcal{C}}_i$ is equivalent to the consideration of $\widehat{\mathcal{C}}_i$.*

**Proposition 3.** *We have SE(3)-invariant density $p_\theta(\widehat{\mathcal{C}}^{(0)})$ if the denoising model $\epsilon_\theta(\mathcal{Z}^{(t)}, t)$ is SE(3)-invariant, i.e., $\epsilon_\theta(\mathcal{Z}^{(t)}, t) = \epsilon_\theta(T_g \mathcal{Z}^{(t)}, t)$, where $T_g$ is some roto-translational transformations of a group element $g \in SE(3)$.*

**Proposition 4.** *The proposed SE(3)-invariant feature embedding $\mathcal{H}(\mathcal{Z}^{(t)}, t)$ ensures that the denoising model $\epsilon_\theta(\mathcal{Z}^{(t)}, t)$ satisfies SE(3)-invariant, provided that its constituent modules preserve permutation equivariance.*

**Proposition 5.** *Our proposed Global MSA, Piece Masked MSA, and Corner Connection Masked MSA always satisfy permutation equivariance.*

Considering Propositions 1 to 5 together, the proposed model generates an SE(3)-invariant density function for the entire puzzle. All proofs are provided in the appendices.

## 6 EXPERIMENTAL RESULTS

To validate our method, we conduct experiments in both 2D and 3D domains. Additionally, to prove the performance contributions of each individual component, we carry out ablation studies, thereby verifying their effectiveness. Here we enumerate our individual components: shape canonicalization (denoted as *SC*), anchor centering (denoted as *AC*), anchor piece-related components (the largest piece selection (denoted as *AO*) and use of anchor($i$) (denoted as *AI*)), connection-wise masked

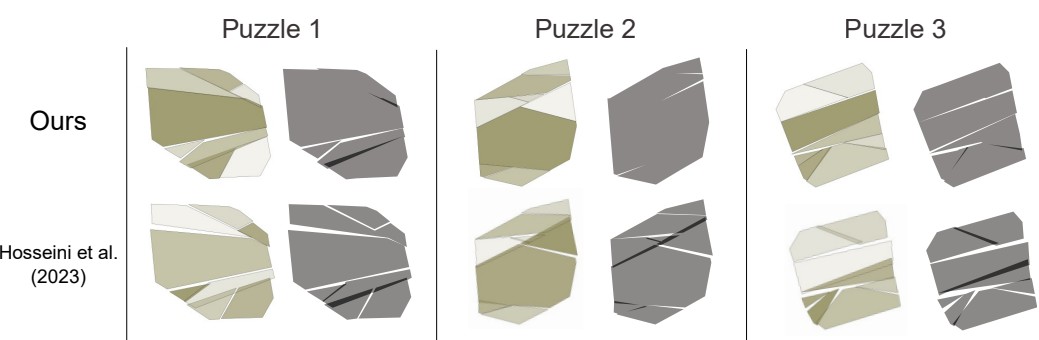

Figure 4: Qualitative results for the 2D recursive-cut dataset.

Table 1: Performance comparison on the 2D cross-cut and recursive-cut puzzles.

| Dataset | Method | Overlap (↑) | Precision (↑) | Recall (↑) |
|---|---|---|---|---|
| Cross-cut puzzle | Harel et al. (2021) | 0.9145 | 0.9562 | **0.9908** |
| | Hosseini et al. (2023) | 0.9410 | 0.9694 | 0.9170 |
| | Ours | **0.9709** | **0.9823** | 0.9666 |
| Recursive-cut puzzle | Hosseini et al.(2023) | 0.9251 | 0.9286 | 0.8950 |
| | Ours | **0.9504** | **0.9679** | **0.9244** |

Table 2: Ablation studies on the respective components for the 2D cross-cut dataset.

| SC | AC | CA | PA | CC | AI | AO | Overlap (↑) | Precision (↑) | Recall (↑) |
|---|---|---|---|---|---|---|---|---|---|
| ✓ | ✓ | ✓ | ✓ | ✓ | ✓ | ✓ | **0.9709** | 0.9823 | 0.9666 |
| | ✓ | ✓ | ✓ | ✓ | ✓ | ✓ | 0.9644 | 0.9817 | 0.9501 |
| ✓ | | ✓ | ✓ | ✓ | | | 0.9572 | 0.9805 | 0.9646 |
| ✓ | ✓ | | ✓ | ✓ | ✓ | ✓ | 0.9682 | 0.9818 | **0.9670** |
| ✓ | ✓ | ✓ | | ✓ | ✓ | ✓ | 0.9336 | 0.9358 | 0.9211 |
| ✓ | ✓ | ✓ | ✓ | | ✓ | ✓ | 0.9701 | **0.9837** | 0.9715 |
| ✓ | ✓ | ✓ | ✓ | ✓ | | ✓ | 0.9632 | 0.9740 | 0.9238 |
| ✓ | ✓ | ✓ | ✓ | ✓ | ✓ | | 0.9634 | 0.9758 | 0.9439 |
| ✓ | ✓ | ✓ | ✓ | ✓ | | | 0.9599 | 0.9734 | 0.9575 |
| | | | ✓ | | | | 0.9215 | 0.9284 | 0.8968 |

attention (denoted as *CA*), piece-wise masked attention (denoted as *PA*), and corner connectivity (denoted as *CC*). Additionally, in the 3D domain, since our dataset generation process enables us to adjust the level of assembled shape complexity, we compare the levels of shape complexity in the ablation study; see the appendices. Each complexity level corresponds to the average number of corners that are included in a target shape. Missing details on training details, datasets, baseline methods, and evaluation metrics can be found in the appendices.

**Analysis on the 2D Puzzles.** 2D model design is essentially similar to its 3D version described in Section 4, but the length of the vectors required to represent all $\mathbf{a}_i$ significantly decreases. This makes all processes noticeably faster compared to the 3D puzzles. Nevertheless, our method generally outperforms the existing methods as shown in Table 1, and qualitative results are available in Figures 3 and 4. In particular, the consistent performance gains are demonstrated on our own 2D recursive-cut puzzle dataset.

The ablation study discussed in Table 2 shows performance loss when the certain option is removed. A notable difference from the 3D results is the relatively minor impact of the CA component. This observation may be attributed to the inherently simpler structural features of 2D shapes compared to 3D puzzles, because each point is connected only to two other points implying an adjacent corner index $j$ is basically adjacent. Supporting this observation, the presence or absence of CC, which enhances information on connected corners, also does not significantly impact the performance.

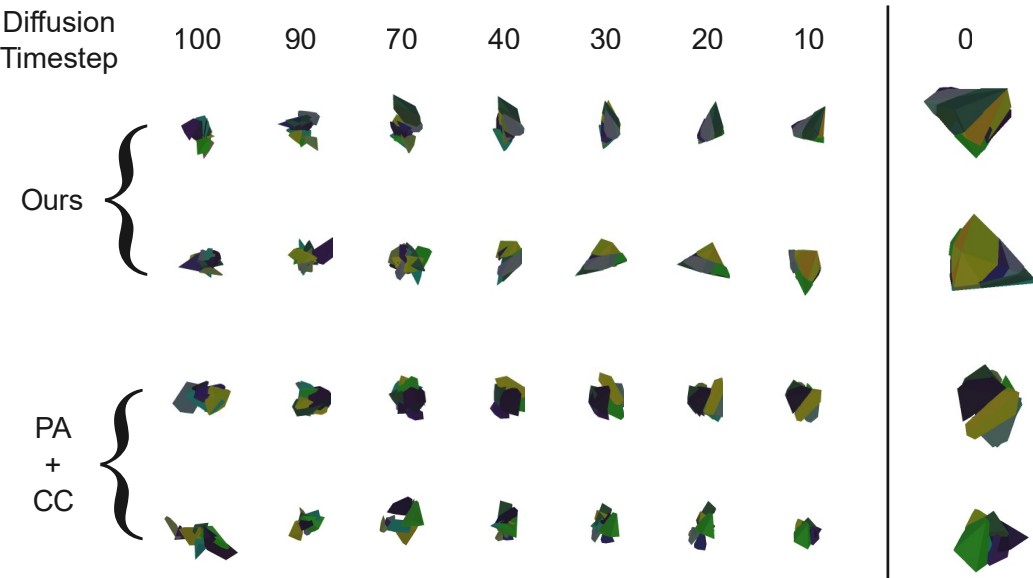

Figure 5: Qualitative results for the 3D shape puzzles.

Table 3: Performance comparison on the 3D recursive-cut puzzles.

| Method | PC ($\uparrow$) | PC-C ($\downarrow$) | RMSE-T ($\downarrow$) | RMSE-R ($\downarrow$) |
|---|---|---|---|---|
| PA, CC | 0.7202 | 0.01436 | 0.9186 | 0.06170 |
| SC, CA, PA, CC | 0.9669 | 0.002398 | 0.3145 | 0.1149 |
| Ours | **0.9955** | **0.001256** | **0.2105** | **0.09284** |

**Analysis on the 3D Puzzles.** We establish a baseline setup, and then demonstrate performance improvements when various components are applied to this baseline. We compare this baseline against a model with additional components applied and the best model with all components. Since PA and CC are not related to explicit equivariance components, applying additional components to this model can be considered as incorporating equivariance. Comparison between this structured baseline and other models is shown in Table 3, and qualitative results are available in Figure 5. We observe that models with additional components significantly outperform the predefined baseline model, showcasing the effectiveness of our components.

As presented in Table 4, we observe intuitive experimental results. We conduct performance evaluations for the models trained by excluding each component from a setup with every component. We confirm that each component plays a significant role in enhancing performance. Additionally, models with a small number of components often result in poor performance. Among components, SC, PA, and CC are most effective. It is noteworthy that three components, SC, CA, and CC, have a significantly greater impact on performance in 3D than in 2D. Their common characteristic is the relation to the piece shapes in 3D space.

**Analysis on the Breaking Bad Dataset.** We also perform analysis on a real-world dataset in order to show that our method can successfully assemble more realistic objects. The Breaking Bad dataset (Sellán et al., 2022) is used for this analysis.

As presented in Table 5, we compare our method to baseline methods. By using the mesh decimation method provided by Open3D (Zhou et al., 2018), we use the sub-sampled version of the meshes from the Breaking Bad dataset (Sellán et al., 2022), to handle the analysis in reasonable time. Note that the performance shown in Table 5 is analyzed by our best performing method with SC, AC, CA, PA, CC, AI, and AO.

As a result, our model clearly mark the best among several diffusion-based assembly approaches. Naturally, feature-based matching methods show better performance but it is slower than the

Table 4: Ablation study on the components in our framework for the 3D dataset.

| DC | SC | AC | CA | PA | CC | AI | AO | PC ($<$ 1e-2) | PC-C | RMSE-R | RMSE-T |
|---|---|---|---|---|---|---|---|---|---|---|---|
| 15 | ✓ | ✓ | ✓ | ✓ | ✓ | ✓ | ✓ | **0.9955** | **0.001256** | **0.2105** | 0.09284 |
| 15 |  | ✓ | ✓ | ✓ | ✓ | ✓ | ✓ | 0.8518 | 0.008291 | 0.8076 | 0.07911 |
| 15 | ✓ |  | ✓ | ✓ | ✓ |  |  | 0.9671 | 0.003039 | 0.2433 | **0.04280** |
| 15 | ✓ | ✓ |  | ✓ | ✓ | ✓ | ✓ | 0.9857 | 0.002417 | 0.2158 | 0.07534 |
| 15 | ✓ | ✓ | ✓ |  | ✓ | ✓ | ✓ | 0.9390 | 0.005063 | 0.2264 | 0.07332 |
| 15 | ✓ | ✓ | ✓ | ✓ |  | ✓ | ✓ | 0.9502 | 0.005537 | 0.2234 | 0.09406 |
| 15 | ✓ | ✓ | ✓ | ✓ | ✓ |  | ✓ | 0.9715 | 0.002677 | 0.2418 | 0.1028 |
| 15 | ✓ | ✓ | ✓ | ✓ | ✓ | ✓ |  | 0.9846 | 0.002154 | 0.2140 | 0.07314 |
| 15 | ✓ | ✓ | ✓ | ✓ | ✓ |  |  | 0.9704 | 0.002427 | 0.2275 | 0.06024 |
| 15 |  |  |  | ✓ | ✓ |  |  | 0.7202 | 0.01436 | 0.8135 | 0.06170 |
| 15 |  | ✓ |  | ✓ |  |  |  | 0.3418 | 0.2792 | 0.9186 | 0.3349 |
| 15 |  |  |  | ✓ |  |  |  | 0.0532 | 0.8536 | 0.8760 | 0.2980 |

Class Pot      Class Ring

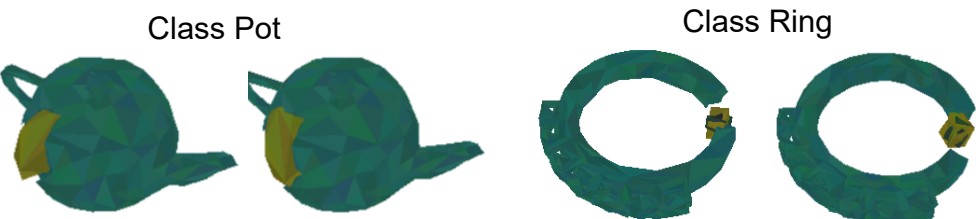

Figure 6: Qualitative results for the Breaking Bad dataset.

Table 5: Performance comparison on the Breaking Bad dataset.

| Category | Method | RMSE-R ($\downarrow$) | RMSE-T ($\downarrow$) | PA (%) ($\uparrow$) |
|---|---|---|---|---|
| Feature-based Matching | Jigsaw | 42.3 | 10.7 | 57.3 |
|  | PuzzleFusion++ | 38.1 | 8.04 | 70.6 |
| Diffusion | Global | 80.7 | 15.1 | 24.6 |
|  | LSTM | 84.2 | 16.2 | 22.7 |
|  | DGL | 79.4 | 15.0 | 31.0 |
|  | SE(3)-Equiv | 79.3 | 16.9 | 8.41 |
|  | DiffAssemble | 73.3 | 14.8 | 27.5 |
|  | Ours | 58.8 | 13.3 | 40.4 |

diffusion-based assembly methods including our method. If the number of pieces in an object increases, feature-based matching methods are likely to fail in the production of assembly results due to memory and compute requirements. Qualitative results can be seen in Figure 6, showing that our model is capable of understanding delicate geometric cues of the Breaking Bad dataset.

## 7 CONCLUSION

In this study, we successfully applied diffusion models in 2D and 3D shape puzzles including the Breaking Bad dataset. We proposed several features that make shape representations invariant across both 2D and 3D environments. Moreover, we addressed the limitations of the traditional assembly metrics and introduced a novel evaluation metric, point connectivity. Finally, the theoretical and empirical results are demonstrated to represent the superiority of our method. In particular, our method shows consistent performance improvements across varying dimensions and conditions.

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

## A  ADDITIONAL DISCUSSION ON RELATED WORK AND PRELIMINARIES

**Shape and Puzzle Assembly Problems.**  In the recent study (Shabani et al., 2023), diffusion models for floorplan generation have been introduced. Using the similar approach, PuzzleFusion (Hosseini et al., 2023) tackles 2D shape puzzle tasks formulating it as a conditional generation problem. The authors of the work (Hosseini et al., 2023) have prepared diverse 2D puzzle datasets including the 2D Voronoi jigsaw and cross-cut datasets. However, this approach exhibits dependency on specific dataset constraints and faces performance degradation under simple permutation or rotation attacks.

**Diffusion Models.**  Diffusion models, in this paper particularly denoising diffusion probabilistic models (Ho et al., 2020), have emerged as one of powerful generative models, since it is capable of generating high-quality samples in various domains. A diffusion model generally consists of two phases: forward and reverse processes. In the forward process, the model adds Gaussian noise to an input every step, effectively transforming an original data distribution into a noise distribution. This gradual process diffuses an initial data $\mathbf{x}_0$ until it becomes an indistinguishable noisy data $\mathbf{x}_T$. Formerly, it is expressed as

$$\mathbf{x}_t = \sqrt{\alpha_t}\mathbf{x}_{t-1} + \sqrt{1-\alpha_t}\boldsymbol{\epsilon}_t, \tag{11}$$

where $\mathbf{x}_t$ is a data at step $t$, $\alpha_t$ is a coefficient that controls the amount of noise added at step $t$, and $\boldsymbol{\epsilon}_t$ is sampled from the multi-variate standard normal distribution $\mathcal{N}(\mathbf{0}, \mathbf{I})$. Note that the coefficient $\alpha_t$ is determined by following a predefined schedule policy, which decreases it as $t$ increases in order to gradually add more noise to the data. Conversely, the reverse process involves reconstructing a cleaner data $\mathbf{x}_{t-1}$ from a noisier data $\mathbf{x}_t$. This allows us to iteratively transform a noisy data $\mathbf{x}_T$ to an original data $\mathbf{x}_0$. A simplified reverse process can be written as follows:

$$\mathbf{x}_{t-1} = \frac{1}{\sqrt{1-\beta_t}}\left(\mathbf{x}_t - \frac{\beta_t}{\sqrt{1-\bar{\alpha}_t}}\boldsymbol{\epsilon}_\theta(\mathbf{x}_t, t)\right), \tag{12}$$

where $\mathbf{x}_{t-1}$ is a denoised estimate at step $t-1$, $\beta_t$ is the variance of the noise added at step $t$, $\bar{\alpha}_t$ is the cumulative product of $1 - \beta_t$ up to step $t$, and $\boldsymbol{\epsilon}_\theta(\mathbf{x}_t, t)$ is the estimated noise predicted by a model with parameters $\theta$ at step $t$.

## B  PROOF OF PROPOSITION 1

*Proof.* Let the set of canonicalized elements be denoted as $\{\mathbf{c}_j\}_{j=1}^m$. Any arbitrary element $\mathcal{C}$ can be written as $\{\mathbf{R}\mathbf{c}_j + \mathbf{t}\}_{j=1}^m$. Transforming an arbitrary $\mathcal{C}_i$ into its canonicalized form $\{\mathbf{c}_j\}_{j=1}^m$ is equivalent to removing the effects of $\mathbf{R}$ and $\mathbf{t}$. According to Section 4.3, setting $t$ as the center of mass and $\mathbf{R}$ as $\mathbf{US}$ from the result of SVD from the result of SVD makes it possible to canonicalize the shape. $\square$

## C  PROOF OF PROPOSITION 2

*Proof.* Let the completed puzzle be denoted as $\widehat{\mathcal{C}} = [\widehat{\mathbf{c}}_1, \widehat{\mathbf{c}}_2, \ldots, \widehat{\mathbf{c}}_m] \in \mathbb{R}^{3 \times m}$. If $\mathcal{C}$ is canonicalized shape, the completed puzzle has a linear relationship with the arrangement parameters, as shown in Equation 1 of Section 3.2.

$$\widehat{\mathcal{C}} = \mathbf{R}_i^{(y)}\mathbf{R}_i^{(p)}\mathbf{R}_i^{(r)}\mathcal{C} + \mathbf{t}, \tag{13}$$

where rotation matrices $\mathbf{R}^{(r)}, \mathbf{R}^{(p)}, \mathbf{R}^{(y)}$ and translation $\mathbf{t}$ are the corresponding transformations of arrangement parameter $\mathbf{a}$. $\square$

## D  PROOF OF PROPOSITION 3

*Proof.* By Proposition 1, we can assume that $\widehat{\mathcal{C}}^{(t)}$ is generated from shape-canonicalized pieces. Proposition 2 allows us to convert the discussion about $\widehat{\mathcal{C}}^{(t)}$ into one about the arrangement parameters $\mathbf{a}^{(t)}$. Meanwhile, it has been proven that the density of the entire assembly remains invariant as long as the diffusion kernel is made equivariant to SE(3) transformation (Xu et al., 2022). If our diffusion model $\boldsymbol{\epsilon}_\theta(\mathcal{Z}^{(t)}, t)$ satisfies SE(3)-invariance, the same property can be shown to hold for the

arrangement parameters $\mathbf{a}^{(t)}$. Let $T_g$ be a roto-translational transformation of a group element $g \in$ SE(3), and let $p(\mathbf{a}^{(T)})$ represent the SE(3)-invariant density, i.e., $p(\mathbf{a}_T) = p(T_g(\mathbf{a}_T))$. If the Markov transitions $p(\mathbf{a}^{(t-1)}|\mathbf{a}^{(t)}$ parameterized by $\epsilon_\theta(\mathcal{Z}^{(t)}, t)$ are also SE(3)-invariant, i.e., $p(\mathbf{a}^{(t-1)}|\mathbf{a}^{(t)}) = p(T_g(\mathbf{a}^{(t-1)})|T_g(\mathbf{a}^{(t)}))$, then the overall density $p_\theta(\mathbf{a}^{(0)}) = \int p(\mathbf{a}^{(T)})p_\theta(\mathbf{a}^{(0:T-1)}|\mathbf{a}^{(T)})\mathrm{d}\mathbf{a}^{(1:T)}$ will also be SE(3)-invariant, as demonstrated by Xu et al. (2022).

$$
\begin{aligned}
p_\theta(T_g(\mathbf{a}^{(0)})) &= \int p(T_g(\mathbf{a}^{(T)}))p_\theta(T_g(\mathbf{a}^{(0:T-1)})|T_g(\mathbf{a}^{(T)}))\mathrm{d}\mathbf{a}^{(1:T)} \\
&= \int p(T_g(\mathbf{a}^{(T)})) \prod_{t=1}^{T} p_\theta(T_g(\mathbf{a}^{(t-1)})|T_g(\mathbf{a}^{(t)}))\mathrm{d}\mathbf{a}^{(1:T)} \\
&= \int p(\mathbf{a}^{(T)}) \prod_{t=1}^{T} p_\theta(T_g(\mathbf{a}^{(t-1)})|T_g(\mathbf{a}^{(t)}))\mathrm{d}\mathbf{a}^{(1:T)} \quad \text{(invariant prior } p(\mathbf{a}^{(T)})) \\
&= \int p(\mathbf{a}^{(T)}) \prod_{t=1}^{T} p_\theta(\mathbf{a}^{(t-1)}|\mathbf{a}^{(t)})\mathrm{d}\mathbf{a}^{(1:T)} \quad \text{(invariant kernels } p(\mathbf{a}^{(t-1)}|\mathbf{a}^{(t)})) \\
&= p_\theta(\mathbf{a}^{(0)}).
\end{aligned}
\tag{14}
$$

$\square$

While this proof closely follows the structure used in the previous work (Xu et al., 2022), the novelty lies in extending the invariance to the arrangement parameters, after fixing the canonicalized pieces. By canonicalizing the puzzle pieces and focusing on the arrangement parameters, we ensure that the density of the entire puzzle assembly is invariant. This division between the canonicalized pieces and arrangement parameters of assembled puzzle introduces a new layer of generalization, which allows us to more efficiently learn the puzzle assembly process while maintaining SE(3)-invariance.

## E    PROOF OF PROPOSITION 4

*Proof.* Let the denoising model $\epsilon_\theta(\mathcal{Z}^{(t)}, t)$ be composed of an SE(3)-invariant feature embedding layer $\mathcal{H}(\mathcal{Z}^{(t)}, t)$ and $L$ permutation-equivariant layers $\mathcal{G}_k, k \in [L]$. Consider an input that undergoes both an SE(3) transformation and a permutation. The resulting output can be expressed as follows:

$$
\begin{aligned}
\epsilon_\theta(\mathcal{Z}^{(t)}, t) &= \mathcal{G}_L(\ldots \mathcal{G}_1(\mathcal{H}(\mathbf{P}T_g\mathcal{Z}^{(t)}, t))) \\
&= \mathcal{G}_L(\ldots \mathcal{G}_1(\mathbf{P}\mathcal{H}(\mathcal{Z}^{(t)}, t))) \quad \text{(SE(3)-invariance } \mathcal{H}(\mathbf{P}T_g\mathcal{Z}^{(t)}, t) = \mathbf{P}\mathcal{H}(\mathcal{Z}^{(t)}, t))) \\
&= \mathbf{P}\mathcal{G}_L(\ldots \mathcal{G}_1(\mathcal{H}(\mathcal{Z}^{(t)}, t))). \quad \text{(Permutation equivariance } \mathcal{G}_k(\mathbf{P}X) = \mathbf{P}\mathcal{G}_k(X))
\end{aligned}
\tag{15}
$$

Thus, the output of the denoising model is permutation-equivariant and SE(3)-invariant. This follows directly from the fact that the invariant embedding layer $\mathcal{H}$ is applied first, ensuring that the SE(3) transformation is absorbed before the permutation-equivariant layers process the input. $\square$

## F    PROOF OF PROPOSITION 5

*Proof.* To demonstrate that MHA is permutation equivariant, let us first define the self-attention operation in the context of a set input $\mathcal{Z} = \{z_1, z_2, \ldots, z_N\}$. The self-attention mechanism computes the attention scores as follows:

$$
\text{Attention}(Q, K, V) = \text{Softmax}(QK^T)V,
\tag{16}
$$

where $Q, K, V \in \mathbb{R}^{N \times d}$ represent the query, key, and value matrices, respectively, each of which depends on the input set $\mathcal{Z}$. Since the attention mechanism operates on the pairwise relationships between elements, applying a permutation matrix $\mathbf{P} \in \mathbb{R}^{N \times N}$ to $\mathcal{Z}$ permutes both $Q$, $K$, and $V$. The result is that the attention scores are permuted consistently, leading to permutation equivariance in the output:

$$
\text{Attention}(\mathbf{P}Q, \mathbf{P}K, \mathbf{P}V) = \mathbf{P}\text{Attention}(Q, K, V).
\tag{17}
$$

Now consider the proposed module, where a permutation-equivariant mask $M \in \mathbb{R}^{N \times N}$ is applied to the attention scores. After computing the attention scores as $\text{Softmax}(QK^T)$, the mask is applied element-wise:

$$\text{MaskedAttention}(Q, K, V, M) = \text{Softmax}(QK^T) \odot MV, \tag{18}$$

where $\odot$ denotes element-wise multiplication. To verify that permutation equivariance is preserved, let the input set $\mathcal{Z}$ be permuted by $\mathbf{P}$. The masked attention mechanism then becomes:

$$\text{MaskedAttention}(\mathbf{P}Q, \mathbf{P}K, \mathbf{P}V, \mathbf{P}M) = \text{Softmax}(\mathbf{P}Q(\mathbf{P}K)^T) \odot \mathbf{P}M\mathbf{P}V$$
$$= \mathbf{P}\left(\text{Softmax}(QK^T) \odot MV\right). \tag{19}$$

Thus, the output of the masked attention remains equivariant under permutation of the input. Since both the MSA operation and the applied mask respect permutation, the overall module preserves permutation equivariance.

$\square$

## G    8-PIECE RECURSIVE-CUT 3D PUZZLES

Unlike the previous work focuses on 2D geometric shape puzzle (Lee et al., 2022; Hosseini et al., 2023), this research explores the shape puzzles in a 3D domain. To achieve this, we propose a novel method to generate a dataset of 3D puzzles. An initial step requires creating random 3D convex meshes. Their shape complexity can be adjusted by the number of points for 3D meshes; this analysis will be shown in Section 6. This step is accomplished through a process of random point sampling, which is combined with the application of the 3D convex hull algorithm (Khosravani et al., 2013). As the subsequent step, these randomly-generated 3D meshes are recursively cut 3 times, yielding 8 irregular pieces. As a result, a set of 8 geometric pieces, derived from a random 3D mesh, forms a single puzzle problem. Leveraging this procedure, we generate a dataset of 30,000 puzzle sets.

The existing work (Lee et al., 2022; Hosseini et al., 2023) on 2D puzzle tasks has utilized datasets with substantial geometric constraints, which are prone to relying on particular models. For instance, the cross-cut dataset mainly solved in PuzzleFusion (Hosseini et al., 2023) comes with 180° matching and edge matching constraints, which serve as hints to assembly models, making the problems much easier. On the contrary, our 3D recursive-cut puzzle dataset eliminates the edge matching constraints that are uncommon in general real-world puzzle tasks. This design choice lets the conventional puzzle assembly methods which relies heavily on these constraints (Lee et al., 2022; Gallagher, 2012) fail to solve the puzzles, and introduces a unique challenge such that more delicate understanding of spatial and geometric relationships between puzzle pieces is required.

## H    TRAINING DETAILS

Diffusion models for 3D:

- Number of parameter: 6,557,449
- Training from scratch using 1 NVIDIA TITAN RTX
- Batch size: 128
- Learning rate: 1e-3
- Optimization technique: AdamWR (Loshchilov & Hutter, 2017). With AdamW optimizer and additional learning rate annealing applied for each 100,000 training step (annealing factor: 0.1).

Diffusion models for 2D:

- Number of parameter: 4984708
- Training from scratch using 1 NVIDIA TITAN RTX
- Batch size: 512
- Learning rate: 1e-3

- Optimization technique: AdamWR (Loshchilov & Hutter, 2017). With AdamW optimizer and additional learning rate annealing applied for each 100,000 training step (annealing factor: 0.1).

For the fair comparison and analysis, all training details excepts the aforementioned options (e.g., SC and AC) remain same throughout the ablation studies.

## I   DATASETS AND BASELINES

In shape puzzles, the following geometric constraints are commonly included.

- 180° matching constraint: This stipulates that the sum of the two angles adjacent to any corner must be equal to 180 degrees, ensuring that the pieces fit together to form straight lines or correct angles.
- Edge matching constraint: This requires that two adjacent edges not only have equal lengths but also align perfectly with one another, ensuring the continuity of the puzzle's surface.
- Fixed target shape: Across a dataset, a ground-truth assembled target shape does not change. While the shape itself is consistent, variations may occur in scale or the target shape may appear rotated in different instances.

Among the datasets used in 2D shape puzzle tasks, one of relatively general 2D shape puzzles is the cross-cut shape puzzle dataset proposed in the work (Hosseini et al., 2023). These puzzles are created by performing $n$ cuts on a randomly-shaped 2D polygon to produce $2^n$ pieces, resulting in a dataset of randomly-partitioned puzzle pieces. While the cross-cut dataset holds the 180° matching and edge matching constraints, our new generalized recursive-cut dataset on 2D space only holds the edge matching constraint, following the similar procedure shown in Section G.

Incorporating the 3D shape puzzle dataset introduced in Section G, we design to apply the step-wise enhancements discussed in Section 4. These enhancements grounded in the principles of equivariance and invariance demonstrate intuitive and substantial performance improvements in both 2D and 3D puzzles.

## J   EVALUATION METRICS

In the 2D shape puzzle tasks, metrics such as overlap score and point connection precision and recall are employed. The overlap score measures the extent to which a reconstructed shape overlaps with a ground-truth shape, providing the indication of both placement accuracy and shape fidelity. The point connection evaluates the model's ability to correctly identify and reconstruct connections between points that are supposed to be connected in the ground-truth.

In the 3D assembly tasks, traditional metrics such as the root mean square errors of rotation (denoted as RMSE-R) and translation (denoted as RMSE-T) are widely used. In addition to this metric, in this research, we propose the use of point connectivity (denoted as PC) as another metric for assessing 3D puzzle performance. This metric evaluates how well the points of interest are connected in the final prediction. In addition to discrete thresholding classification of point connection, we use the continuous values of point distances (denoted as PC-C) for this metric, enabling the more detailed evaluation of assembly accuracy.

## K   PERFORMANCE GAIN ACROSS DIFFERENT DATASET COMPLEXITIES

As shown in Table 6, our method consistently enhances performance across datasets created from assembled shapes of varying complexity. This consistent improvement underscores the effectiveness of introducing equivariance into our model, confirming its ability to enhance performance irrespective of the dataset. A key observation here is that despite the increase in dataset complexity, leading to a longer input query length and reduced complexity in some instances, there is no significant difference in performance. Intriguingly, at complexity 30, the performance was marginally better than those of complexity 15. This suggests that even as shape complexity increases, with the number of pieces cut remaining at 8, the interfaces between pieces, compared to non-joining surfaces

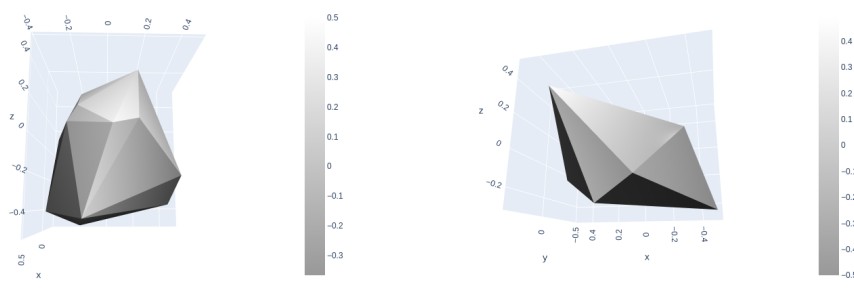

(a) A sample of dataset complexity 15       (b) A sample of dataset complexity 8

Figure 7: Difference of given datasets. Visualization of assembled target shape of puzzle.

Table 6: Ablation study for the 3D dataset, for dataset complexity 8/30. Note that dataset complexity (DC) is indicated by the average number of corners in the assembled shape.

| DC | SC | AC | CA | PA | CC | AI | AO | PC ($< 1e$-2) | PC-C | RMSE-R | RMSE-T |
|---|---|---|---|---|---|---|---|---|---|---|---|
| 8 | ✓ | ✓ | ✓ | ✓ | ✓ | ✓ | ✓ | **0.9947** | **0.001310** | 0.1700 | 0.1104 |
| 8 | ✓ | ✓ | ✓ | ✓ | ✓ | ✓ | | 0.9912 | 0.001497 | **0.1190** | 0.1223 |
| 8 | ✓ | ✓ | ✓ | ✓ | ✓ | | | 0.9726 | 0.001962 | 0.2550 | 0.09332 |
| 8 | | ✓ | ✓ | ✓ | ✓ | | | 0.8257 | 0.01035 | 0.6230 | 0.07110 |
| 8 | | | ✓ | ✓ | ✓ | | | 0.7686 | 0.01386 | 0.7921 | **0.06058** |
| 8 | | | | ✓ | ✓ | | | 0.7180 | 0.005537 | 0.2234 | 0.09406 |
| 30 | ✓ | ✓ | ✓ | ✓ | ✓ | ✓ | ✓ | **0.9977** | **0.0009713** | 0.2200 | 0.09861 |
| 30 | ✓ | ✓ | ✓ | ✓ | ✓ | ✓ | | 0.9945 | 0.001309 | 0.1691 | 0.1029 |
| 30 | ✓ | ✓ | ✓ | ✓ | ✓ | | | 0.9918 | 0.001632 | **0.1182** | 0.0828 |
| 30 | | ✓ | ✓ | ✓ | ✓ | | | 0.8810 | 0.007453 | 0.4546 | 0.09136 |
| 30 | | | ✓ | ✓ | ✓ | | | 0.8256 | 0.01436 | 0.6274 | **0.07597** |
| 30 | | | | ✓ | ✓ | | | 0.7702 | 0.01327 | 0.7910 | 0.09044 |

of pieces, might develop geometric distinctions significant enough to facilitate geometric reasoning. This potentially makes the assembly task easier to solve, despite the overall increase in shape complexity and input query length. Dataset visualization for different dataset complexity can be seen in Figure 1.

## K.1 CONSISTENT PERFORMANCE GAIN

Table 7: Ablation study for the 3D dataset, showing consistent performance improvements when shape canonicalization (SC) is applied.

| DC | SC | AC | CA | PA | CC | AI | AO | PC ($< 1e$-2) | PC-C | RMSE-R | RMSE-T |
|---|---|---|---|---|---|---|---|---|---|---|---|
| 15 | ✓ | ✓ | ✓ | ✓ | ✓ | ✓ | ✓ | 0.9955 | 0.001256 | 0.2105 | 0.09284 |
| 15 | | ✓ | ✓ | ✓ | ✓ | ✓ | ✓ | 0.8518 | 0.008291 | 0.8076 | 0.07911 |
| 15 | ✓ | ✓ | ✓ | ✓ | ✓ | | | 0.9704 | 0.002427 | 0.2275 | 0.06024 |
| 15 | | ✓ | ✓ | ✓ | ✓ | | | 0.9159 | 0.002221 | 0.6165 | 0.04189 |
| 15 | ✓ | | ✓ | ✓ | ✓ | | | 0.9239 | 0.002195 | 0.5463 | 0.08452 |
| 15 | | | ✓ | ✓ | ✓ | | | 0.8960 | 0.003461 | 0.7180 | 0.06254 |
| 15 | ✓ | | | ✓ | ✓ | | | 0.8363 | 0.007180 | 0.8015 | 0.06860 |
| 15 | | | | ✓ | ✓ | | | 0.7202 | 0.01436 | 0.8135 | 0.06170 |

As demonstrated in Table 7, the application of each option, notably Shape Canonicalization (SC), contributes independently and consistently to performance enhancement across various setups. Furthermore, through extensive experimentation with other options, although not included in this supplement, we assert that our method maintains consistent performance improvement regardless of the setup applied.

## L    LIMITATIONS

The model is specifically designed to tackle puzzle datasets, and as such, this method cannot be directly applied to point clouds. Some adjustments around the options will be required to ensure compatibility and effectiveness in a different context.

Training time is significantly influenced by the length of the query, which equates to the overall number of corners in the puzzle dataset. Due to the overcomplete representation, the query length is determined by both the number of pieces and the number of corners per piece. This factor can lead to exceedingly prolonged training times, especially in scenarios where the piece number exceeds 10 or the data complexity is greater than 30.

