# OpenReview forum: "Shape Assembly via Equivariant Diffusion"
_ICLR.cc/2025/Conference — ICLR 2025 Conference Withdrawn Submission_

### Official Review · Reviewer_Ysq8 · 2024-10-24

**Soundness:** 3
**Presentation:** 3
**Contribution:** 3
**Rating:** 6
**Confidence:** 3

**Summary:**

This paper utilizes diffusion models to tackle 2D and 3D shape puzzles problem. It proposes several features that make shape representations invariant across both 2D and 3D environments and provides extensive empirical experiments as well as solid theoretical analysis to prove the effectiveness.

**Strengths:**

1. The motivation of imposing SE(3) invariance and equivariance in feature embedding is straightforward, and the way these constraints are injected is very novel.

2. The theroetical analysis and extensive ablation studies are very solid, indicating the effectiveness of the design.

3. The paper is well written and easy to read.

**Weaknesses:**

1. There are some grammar mistakes and formatting issues in the paper, please polish the writing.

2. Section 4.1 does not give the clear definition of the overcomplete representations $a_{i, j}$, I assume it's the arrangement parameter of each corner point?

3. In Section 4.3 when introducing the anchor centering mechanism, the author does not define the notation $a_{p, 1}$, does it mean the arrangement parameter for the first corner point of anchor piece? Does this anchor remain consistent for all pieces?

**Questions:**

Please refer to previous section.

---

### Official Review · Reviewer_JdhC · 2024-10-27

**Soundness:** 3
**Presentation:** 2
**Contribution:** 2
**Rating:** 3
**Confidence:** 5

**Summary:**

This paper looks at the problem of shape assembly in the setting of geometric assembly in which assembly has to rely on geometric features since appearance features are not present. This is a challenging problem and has practical applications in computer vision, computer graphics and robotics. This paper proposes a method based on diffusion where the process of solving the shape assembly problem can be modeled as a diffusion process. Results show some improvements over the competing methods.

**Strengths:**

1. A new dataset for the task of 3D puzzle is proposed. This is an interesting dataset and could be useful for future research in this direction.

2. Theoretical analysis and empirical results are both provided with additional ablation experiments also included.

3.

**Weaknesses:**

1. In Table 5, the proposed method performs much worse compared to existing methods such as jigsaw. It is unclear what advantages the proposed method has over Jigsaw.

2. The rendering style is a bit confusing. For example, in figure 6 it is unclear how many fractures this example has.

3. It is unclear how the proposed method compares to other methods for the example in figure 6

**Questions:**

Please see the weaknesses above

---

### Official Review · Reviewer_qn4o · 2024-11-03

**Soundness:** 3
**Presentation:** 2
**Contribution:** 2
**Rating:** 5
**Confidence:** 3

**Summary:**

This work addresses the challenge of reassembling randomly partitioned 2D and 3D shapes, relying solely on geometric features in pattern-free, irregular contexts. The authors propose a generative diffusion process that maintains roto-translational equivariance, demonstrating its effectiveness through experiments and ablation studies on various puzzle benchmarks.

**Strengths:**

The problem is challenging and highly ill-posed.
The dataset contribution.

**Weaknesses:**

The model is trained mainly supervised by ground truths, while for this problem, a re-assembling loss should be able to drive self-supervised training. For such supervised networks, the outputs may overfit on training set. The generalization ability is not evaluated.
In Figure 3-4, comparisons on 2D puzzles show improvements by the proposed method are minor.
Colors in Figure 5-6 are hard to recognize, making the figures hard to read.
The method based on feature matching is much better than the proposed one, making me confused about the contributions and improvements. Other than memory and computational costs, feature matching based approaches seem much better.

**Questions:**

See above.

---

### Official Review · Reviewer_oens · 2024-11-04

**Soundness:** 2
**Presentation:** 2
**Contribution:** 2
**Rating:** 5
**Confidence:** 3

**Summary:**

The paper proposes 3D (or 2D) puzzle solver using only geometric information (with no textural information). The proposed method assumes that the piece consists of polygonal faces, straight edges and sharp corner. The puzzle problem is formulated as estimating rotation and translation Euclidean transformation for each corner that minimizes the loss function (MSE of noise and matching). The optimization is done by diffusion model process. The major novel contribution is to propose a layer that generates equivariant feature embedding. The proposed method presents better performance on 2D and 3D shape puzzles than prior works.

**Strengths:**

The major difference from the prior works is to design intermediate layers to make embedding equivariant. And, the performance on 2D and 3D puzzle dataset is better than prior works.

**Weaknesses:**

The reviewer concerns novelty of the paper. The reviewer is unable to understand major differences from Hosseini et al., 2023. The problem formulation, optimization and loss functions are very similar with slight modification. And, the reviewer is unable to find a connection between the equation 4 and equivariant feature embeddings. The explanation of core idea is ambiguous to the reviewer.

Some tentative typos and unclear statements
- L226, “In the sense of puzzle, it is considered …” what is ‘it’ mean? Unclear to understand the meaning of the statement.
- Please verify, Line 85, “f(T(x)) = f(x) or f(T(x)) = T(f(x))”.
- L322, in the equation “m” should be “l”?
- The author did not explain “design a dataset generating framework” specified in the contributions L46.
- The author did not explain “a novel evaluation metric” in detail specified in the Conclusion section.
- There are more ambiguous statements.

**Questions:**

Please elaborate what the difference is from Hosseini et al., 2023
Please elaborate more why equation 4 is equivariant feature embedding?
Why the same shape with different R and T should have the same embedding in L242? The reviewer thinks that the embeddings should be dependent on not only shape but also R and T.

---

### Note · Authors · 2024-11-14

**Comment:**

We deeply appreciate the reviewers' comment.  We will improve our work based on your valuable reviews.

**Withdrawal Confirmation:**

I have read and agree with the venue's withdrawal policy on behalf of myself and my co-authors.